# Multi-Omics Analyses Reveal the Antifungal Mechanism of Phenazine-1-Carboxylic Acid Against *Pseudogymnoascus destructans*

**DOI:** 10.3390/jof12010016

**Published:** 2025-12-25

**Authors:** Zihao Huang, Shaopeng Sun, Zhouyu Jin, Yantong Ji, Jiaqi Lu, Ting Xu, Keping Sun, Zhongle Li, Jiang Feng

**Affiliations:** 1College of Life Science, Jilin Agricultural University, Changchun 130118, China; mappil3456@163.com (Z.H.);; 2Jilin Provincial Key Laboratory of Animal Resource Conservation and Utilization, Northeast Normal University, Changchun 130117, China; 3Jilin Provincial International Cooperation Key Laboratory for Biological Control of Agricultural Pests, Changchun 130118, China

**Keywords:** *Pseudogymnoascus destructans*, phenazine-1-carboxylic acid, antifungal mechanism, multi-omics, bat

## Abstract

White-nose syndrome (WNS) is an infectious disease of bats caused by the psychrophilic fungus *Pseudogymnoascus destructans*. Phenazine-1-carboxylic acid (PCA) is a microbial secondary metabolite with broad-spectrum antifungal activity. Previous studies show that PCA suppresses the growth of *P. destructans* at low concentrations, yet its mechanism remains unclear. Here, we evaluated the in vitro antifungal activity of PCA. We then investigated its potential mechanism using physiological and biochemical assays, as well as integrated transcriptomic and metabolomic analyses. PCA showed effective antifungal activity against *P. destructans* (EC_50_ = 32.9 μg/mL). Physiological and biochemical assays indicated that PCA perturbed cell wall organization and increased membrane permeability, leading to leakage of intracellular contents. It also induced oxidative stress, DNA damage, and apoptosis. Multi-omics integration revealed that PCA markedly perturbed cell wall and membrane metabolism, virulence factor expression, and energy metabolism. It provoked oxidative stress while downregulating genes involved in the cell cycle, DNA replication, and repair. Together, these findings delineate the inhibitory effects of PCA on *P. destructans* in vitro, provide initial mechanistic insights into its antifungal action, and suggest that PCA merits further evaluation as a possible component of environmentally compatible strategies for WNS management.

## 1. Introduction

In recent years, pathogenic fungi have caused unprecedented population declines and extinctions in animals and plants, with profound consequences for ecosystem functioning and human health [1]. White-nose syndrome (WNS), first reported in New York, USA, in 2006, spread rapidly across North America and now threatens bat diversity and the stability of cave ecosystems [2]. The causative agent, the psychrophilic fungus *Pseudogymnoascus destructans*, persists for long periods in soils and cave substrates, creating an environmental reservoir that makes field management highly challenging [3]. At present, there is no effective treatment for WNS. Existing chemical and biological approaches are limited by efficacy, safety, tolerability, and ecological risks, underscoring the urgent need for antifungal candidates with defined mechanisms and reliable, environmentally compatible sources.

Interestingly, the epidemiology of WNS varies markedly among regions. In China, *P. destructans* has been detected, yet large-scale mortality has not been observed, and fungal loads are lower than in North America [4,5]. This pattern may reflect long-term coevolution between host and pathogen or naturally occurring host resistance mechanisms, which remain poorly understood [6]. A systematic study of epidemiological features in China may inform management strategies in North America. Notably, *Pseudomonas yamanorum* GZD14026 isolated from bat skin in China strongly inhibits *P. destructans*, and the active compound was identified as phenazine-1-carboxylic acid (PCA) [7]. Whole-genome sequencing of this strain revealed a PCA biosynthetic gene cluster, suggesting that bat-associated microbes may contribute to natural defense against WNS by secreting PCA. A promising direction for WNS control is to deploy microbial metabolites to suppress *P. destructans* growth [8]. PCA is produced by diverse microbes, including *Pseudomonas* and *Streptomyces*, shows broad-spectrum antibiotic properties, and is registered in China as a biocontrol fungicide [9,10]. It inhibits multiple plant pathogenic fungi and bacteria. For example, PCA effectively suppresses *Pestalotiopsis kenyana*, the causal agent of bayberry leaf lesions [9], and strongly inhibits several fungal pathogens associated with grapevine trunk diseases [11]. It also shows broad potential for controlling *Botrytis cinerea* and inhibiting exopolysaccharide synthesis [12]. These data demonstrate that PCA combines efficacy, relatively low toxicity, and environmental compatibility, making it an attractive candidate for practical antifungal applications [9]. However, despite this growing body of work, the cellular and molecular mechanisms underlying PCA’s activity against *P. destructans* remain unelucidated. Specifically, the potential contribution of bat-associated, PCA-producing bacteria to regional variations in WNS impact has not been systematically examined. Furthermore, multi-omics approaches have yet to be integrated to dissect the mode of action of PCA against *P. destructans*.

Mechanistic studies in several plant-pathogenic fungi have shown that PCA inhibits fungal growth through multiple processes. In species such as *P. kenyana* and *Valsa mali*, PCA acts as a redox-active phenazine that disrupts cellular respiration and redox homeostasis, leading to accumulation of reactive oxygen species (ROS) and a reduction in catalase and superoxide dismutase activities [13]. Exposure to PCA commonly causes abnormal morphology, membrane damage, and a decrease in mitochondrial membrane potential, and transcriptomic analyses in plant pathogens have revealed strong responses of redox-related genes [9]. Beyond stress-related effects, PCA may interact with key metabolic pathways; for instance, in *V. mali*, PCA has been shown to interfere with the glyoxylate cycle and lipid utilization by affecting isocitrate lyase activity [14]. Whether similar or distinct pathways are engaged in *P. destructans* remains completely unknown.

Against this background, we conducted in vitro experiments that integrated physiological and biochemical assays with transcriptomic and metabolomic analyses to delineate how PCA inhibits *P. destructans*. The objectives were to: (1) assess cell wall and membrane integrity, redox status, and changes in energy metabolism at the phenotypic and cellular levels; and (2) identify affected molecules and metabolic pathways and perform integrative analyses to propose a comprehensive mechanistic model of PCA action based on phenotypic and multi-omics evidence. To our knowledge, this is the first study to systematically dissect the mode of action of PCA against *P. destructans* and the first to integrate targeted physiological and biochemical assays with transcriptomics and metabolomics for this pathogen. These insights will inform the design of effective interventions and provide a mechanistic basis for the environmentally friendly control of WNS using PCA.

## 2. Materials and Methods

### 2.1. Strain and Reagents

*P. destructans* strain JHCN111a was preserved by the Jilin Provincial International Cooperation Key Laboratory for Biological Control of Agricultural Pests (Changchun, China). Phenazine-1-carboxylic acid (PCA, CAS:2538-68-3, purity 98%) was purchased from Macklin Biochemical Technology Co., Ltd. (Shanghai, China). Fungal Fluorescence Stain Kit (CFW Method) was obtained from Solarbio Science and Technology Co., Ltd. (Beijing, China). Reactive Oxygen Species Assay Kit, ATP Content Assay Kit, Superoxide Dismutase Activity Assay Kit and Catalase Activity Assay Kit were sourced from Box Biotechnology Co., Ltd. (Beijing, China). DAPI Staining Solution and Annexin V-FITC Apoptosis Detection Kit were purchased from Beyotime Biotechnology Co., Ltd. (Shanghai, China).

### 2.2. Revival of P. destructans and In Vitro Antifungal Assays

The glycerol spore stock was removed from −80 °C storage and thawed at room temperature. A 100 μL aliquot was evenly spread onto 90 mm Sabouraud dextrose agar (SDA) plates. Plates were incubated at 13 °C and 85% relative humidity (RH) for 14 days to revive the strain. To collect spores, 10 mL of 1× PBST (PBS with 0.05% Tween 20, *v*/*v*) was added to the plate surface. The colony surface was gently scraped with a sterile inoculating loop and mixed. The suspension was then filtered through sterile gauze to remove hyphal fragments. Spores were counted using a hemocytometer, and the suspension was adjusted with PBST to 2 × 10^5^ spores/mL [7].

Antifungal activity of PCA against *P. destructans* was evaluated on solid Sabouraud dextrose agar (SDA) using an agar dilution assay adapted for slow-growing filamentous fungi. The agar dilution method was chosen primarily to accommodate the slow 14-day growth rate of *P. destructans*. This approach simultaneously mitigated issues inherent in long-term liquid culture, such as evaporation and sedimentation, while facilitating morphological observation [15]. PCA from a DMSO stock was added to autoclaved SDA cooled to about 50 °C to obtain final concentrations of 10, 20, 30, 40, 50, and 60 μg/mL. The final DMSO concentration was 0.5% (*v*/*v*) for all plates. Thirty milliliters of medium were poured into each plate and allowed to solidify at room temperature. Control plates (CK) contained 0.5% DMSO without PCA. Next, 100 μL of the 2 × 10^5^ spores/mL suspension was evenly spread on each plate. Three biological replicates were prepared per concentration. All plates were incubated at 13 °C and 85% RH for 14 days, and colony images for area quantification were acquired at this single endpoint for every treatment and the control. Plates were then photographed under standardized imaging conditions, and total colony area was quantified with Ilastik v1.4.1rc2 [16] and Image J v1.54f [17]. The inhibition rate was calculated as follows: Inhibition (%) = (1 − A_P_/A_C_) × 100%, where A_C_ and A_P_ represent the colony areas of the control and PCA-treated groups, respectively [18]. For all quantitative analyses, including growth inhibition (Figure 1A) and the concentration–response curve and EC_50_ estimation (Figure 1C), the 14-day colony areas were used.

### 2.3. Assessment of Cell Wall and Membrane Integrity

To assess cell wall and plasma membrane integrity, we performed calcofluor white (CFW) and propidium iodide (PI) staining, measured relative electrical conductivity, and quantified extracellular nucleic acid and protein leakage (OD_260_/OD_280_). For CFW and PI staining, mycelia grown on SDA were collected (20 mg per sample), assigned to CK or PCA treatment (30 μg/mL), and washed in PBS two to three times. The CFW working solution was prepared according to the manufacturer’s instructions (CFW:PBS = 1:1, *v*/*v*). One milliliter was added to each sample and incubated in the dark for 30 min. The PI working solution was prepared by mixing 10 μL PI stock with 190 μL PBS (1:20, *v*/*v*). Each sample was incubated in 200 μL of this solution in the dark for 30 min. Cell wall architecture and membrane integrity were examined by confocal laser scanning microscopy (CLSM; Leica, Wetzlar, Germany).

For relative electrical conductivity and OD_260_/OD_280_ measurements, PCA concentrations were set to CK, 20, 40, and 50 μg/mL. All treatments were performed in at least three independent biological replicates. For conductivity assays, 0.1 g mycelia were suspended in 10 mL sterile distilled water containing PCA at the indicated concentrations. Conductivity was recorded at 0, 4, 8, 12, 16, 20, and 24 h (HORIBA, Kyoto, Japan). The 0 h reading was designated L0, each time point as L1, and the value after boiling the mycelia for 30 min as L2. For each sample, L2 was obtained from the same suspension after boiling, so that any effect of PCA on maximal conductivity was normalized within that sample. Relative electrical conductivity was calculated as: Relative electrical conductivity (%) = (L1 − L0)/(L2 − L0) × 100% [9]. Untreated mycelial suspensions without PCA were included in parallel as negative controls. Additionally, to exclude the interference of PCA itself, blank controls containing PCA at corresponding concentrations in sterile distilled water (without mycelia) were measured. The conductivity of these blanks remained stable and negligible over the 24 h period and after boiling (Appendix A). For OD_260_ and OD_280_ assays, washed mycelia (0.1 g) were suspended in PBS containing PCA at 20, 40, or 50 μg/mL. Samples were collected at 0, 2, 4, 6, and 8 h, centrifuged at 12,000× *g* for 5 min, and supernatants were measured at 260 nm and 280 nm with a microplate reader (Thermo Scientific, Waltham, MA, USA) to quantify extracellular leakage of nucleic acids and proteins. For each PCA concentration, a corresponding solution containing PCA in the same PBS but without mycelia was used as the blank when measuring OD_260_ and OD_280_, to exclude any background absorbance attributable to PCA or the medium matrix.

### 2.4. Detection of Oxidative Stress and Energy Metabolism

For catalase (CAT), superoxide dismutase (SOD), and adenosine triphosphate (ATP) assays, *P. destructans* mycelia were exposed to PCA at 20, 40, or 50 μg/mL for 14 days. All treatments were performed in at least three independent biological replicates. For each assay, 0.1 g mycelia were mixed with extraction buffer at a 1:10 (*w*/*v*) ratio, homogenized on ice, and clarified by centrifugation at 10,000× *g* for 20 min at 4 °C. Supernatants were analyzed according to the kit instructions, and values were calculated as specified. For reactive oxygen species (ROS) measurements, mycelia were treated with PCA at 30 μg/mL. Mycelia (20 mg) were suspended in 500 μL PBS containing 10 μM 2′,7′-dichlorodihydrofluorescein diacetate (DCFH-DA) and incubated at 37 °C for 20 min in the dark, and then washed two to three times with PBS. Intracellular ROS fluorescence was examined by CLSM.

### 2.5. Detection of DNA Damage and Apoptosis

Mycelia grown on SDA plates (13 °C, 85% RH, 14 days) were collected. For each sample, 20 mg of mycelia were weighed and assigned to CK or PCA treatment (30 μg/mL). Samples were washed in PBS two to three times and then fixed in 70% ethanol at room temperature for 15 min. Samples were incubated in 500 μL PBS containing 1 μg/mL 4′,6-diamidino-2-phenylindole (DAPI) in the dark for 30 min. After an additional PBS wash, DNA damage was assessed by CLSM. For apoptosis assays, washed samples were resuspended in 200 μL of the kit-supplied Annexin V binding buffer. Five microliters of Annexin V conjugated to fluorescein isothiocyanate (FITC) and 10 μL propidium iodide (PI) were added, and samples were incubated for 20 min at room temperature in the dark. Samples were washed with PBS two to three times and imaged by CLSM.

### 2.6. Transcriptomic Analysis

Mycelial samples of *P. destructans* treated with 30 μg/mL PCA and untreated controls were collected (20 mg per sample, three biological replicates), snap-frozen in liquid nitrogen, and stored at −80 °C. Total RNA was extracted with TRIzol using the Total RNA Extractor kit (Invitrogen, Waltham, MA, USA) [19]. RNA concentration was measured with a Qubit 2.0 fluorometer (Invitrogen, Waltham, MA, USA). Integrity and genomic DNA contamination were assessed by agarose gel electrophoresis [20]. RNA that passed quality checks was used to construct cDNA libraries according to the library preparation kit instructions. Libraries that passed quality control were sequenced on the DNBSEQ T7 platform (BGI, Shenzhen, China) with paired-end reads. The raw data have been deposited in the NCBI Sequence Read Archive under accession PRJNA1250769.

Raw reads were assessed with FastQC v0.12.1 [21] and trimmed with Trimmomatic v0.39 to obtain clean reads [22]. Clean reads were aligned to the reference genome GCF_001641265.1 using HISAT2 v2.2.1, and gene expression was quantified as TPM with StringTie v2.2.1 [23]. Differential expression was analyzed with DESeq2 v1.42.0 [24]. Multiple testing was corrected by the Benjamini–Hochberg method, and genes were considered differentially expressed at FDR < 0.05 with |log_2_FC| > 1. Overall variation among samples was evaluated by principal component analysis (PCA) using the vegan (v2.6-4) package in R v4.3.1 [25]. Gene Ontology (GO) and Kyoto Encyclopedia of Genes and Genomes (KEGG) enrichment analyses were performed with clusterProfiler v4.2.2 (*p* < 0.05), and KEGG pathway enrichment was validated by gene set enrichment analysis (GSEA) [26]. Gene association networks for key pathways were constructed and visualized on the OmicShare tools platform [27].

### 2.7. Real-Time Quantitative Polymerase Chain Reaction (RT-qPCR)

Total RNA extracted for transcriptomic analysis was reverse transcribed into cDNA using StarScript Pro All-in-one RT Mix with gDNA Remover (GeneStar, Nanjing, China). Amplification was performed on a QuantStudio 3 system (Thermo Scientific, USA) using 2× RealStar Fast SYBR qPCR Mix (Low ROX) (GeneStar, China). Each reaction contained 50 μL. Cycling conditions were 95 °C for 2 min, followed by 40 cycles of 95 °C for 5 s and 60 °C for 30 s. *EFG1* served as the internal reference gene, and twelve target genes were assayed. Relative expression levels were calculated using the 2^−ΔΔCt^ method [28]. All primer sequences are listed in Appendix A.

### 2.8. Metabolomics Analysis

Mycelial samples were collected under the same conditions as the transcriptomic assays (CK and 30 μg/mL PCA; *n* = 6) and processed for metabolite extraction and analysis following Zhao et al. [29]. Briefly, thawed samples at 4 °C were mixed with 400 μL prechilled methanol/acetonitrile/water (2:2:1, *v*/*v*), vortexed, held at −20 °C for 20 min, and centrifuged at 14,000× *g* for 20 min at 4 °C. Supernatants were collected and dried under vacuum. For mass spectrometry, dried extracts were reconstituted in 100 μL of 50% acetonitrile in water, vortexed, centrifuged at 14,000× *g* for 15 min at 4 °C, and the supernatants were injected. Analyses were performed on an ultrahigh performance liquid chromatography system (Vanquish UHPLC, Thermo, Waltham, MA, USA) coupled to an Orbitrap mass spectrometer (Q Exactive HF, Thermo). Data were acquired in positive and negative electrospray ionization modes.

Raw data were processed in XCMS for peak detection, retention time correction, and alignment [30]. Metabolites were annotated by matching to MassBank, METLIN, HMDB, MoNA, and KEGG; annotations met Metabolomics Standards Initiative level 2 criteria [31]. Orthogonal partial least squares discriminant analysis (OPLS-DA) was performed in MetaboAnalyst 6.0, and potential overfitting was evaluated with 200 permutation tests [32]. Differential metabolites (DEMs) were defined by variable importance in projection (VIP) > 1 with *p* < 0.05. Pathways affected by PCA were enriched and annotated against KEGG and visualized using the OmicShare tools platform [27].

### 2.9. Integrative Analysis of Transcriptomics and Metabolomics

Based on dispersion in the OPLS-DA score plot, three metabolomics samples were selected for integration with the transcriptomic data. O2PLS models were built in R with the OmicsPLS (v2.0.2) package. The numbers of joint latent variables and orthogonal components were selected by tenfold cross-validation, and model stability was evaluated with 200 permutation tests [33]. Genes and metabolites were ranked by absolute loading on the joint components, and the top twenty variables were used to generate loading plots and joint score plots. At the sample level, Pearson correlations were calculated between gene expression and metabolite abundance. Significant pairs with |*r*| > 0.8 and *p* < 0.05 were retained to construct a network linking genes and metabolites. For pathway analysis, DEGs and DEMs were mapped to KEGG for joint enrichment. Pathways were ranked by the number of shared members, and the top 15 were reported.

### 2.10. Statistical Analysis

All experiments included at least three biological replicates, and data are presented as mean ± SD. Statistical analyses of biochemical and phenotypic data were performed in SPSS 27.0 (IBM, USA). Differences among groups were assessed with one-way analysis of variance (ANOVA), followed by Tukey’s HSD for post hoc comparisons. Statistical significance was set at *p* < 0.05. Figures were prepared in Origin 2024 (OriginLab, USA).

## 3. Result

### 3.1. Antifungal Activity of PCA Against P. destructans

Using mycelial growth-rate measurements, we evaluated the in vitro antifungal activity of PCA against *P. destructans* across 10–60 μg/mL. As shown in Figure 1A,B, PCA inhibited mycelial growth in a concentration-dependent manner. Macroscopic morphological observations (Figure 1B) revealed that increasing PCA concentrations resulted in evident changes: colonies were smaller and more condensed, with the characteristic velvety surface hyphae becoming sparser. This suggests that PCA impedes both overall fungal growth and hyphal branching. At 20 μg/mL, inhibition was 23.64 ± 1.45%; at 40 μg/mL, it increased to 68.42 ± 2.45%; and at 50 μg/mL, it exceeded 87.18 ± 0.93%. The dose–response relationship was fitted with a four-parameter logistic (4PL) model, with maximal inhibition fixed at 100%. The model fit was in excellent agreement (*R*^2^ = 0.98295), yielding an EC_50_ of 32.92 ± 1.43 μg/mL (Figure 1C). This EC_50_ is in close agreement with the IC_50_ of 32.08 μg/mL for PCA against *P. destructans* reported by Li et al. (2022) using a broth assay [7]. This concordance between agar and broth assays conducted under different conditions supports the robustness of our phenotypic measurements. These results indicate that PCA exerts consistent, concentration-dependent inhibitory effects against *P. destructans* in vitro.

### 3.2. Effects of PCA on Cell Wall and Membrane Integrity of P. destructans

CFW staining showed uniform and continuous fluorescence in control hyphae. After PCA treatment, fluorescence intensified and became patchy, and the septal signal weakened, changes that are consistent with alterations in cell wall architecture and possible cell wall stress (Figure 2A). To assess plasma membrane integrity, PI staining was performed. Treated hyphae showed markedly stronger red fluorescence, consistent with increased membrane permeability and membrane disruption (Figure 2B). To further assess changes in membrane permeability, we measured the relative electrical conductivity of the surrounding medium as an indirect indicator. As shown in Figure 2C, conductivity increased over time and with concentration. Changes were minor at the low dose, whereas the medium and high doses exceeded the control within hours. In contrast, conductivity in the “water + PCA” control solutions remained low and changed only slightly over 24 h, and boiling these solutions produced only minor additional increases (Appendix A), indicating that PCA itself has only a minor direct effect on conductivity and that the observed increases mainly reflect electrolyte leakage from fungal cells rather than distortions of L2. We also quantified extracellular leakage of nucleic acids and proteins as indicators of permeability. As shown in Figure 2D,E, both readouts increased over time and with concentration, and by 8 h all treatment groups exceeded the control, further indicating that PCA increases membrane permeability and compromises cell envelope integrity in *P. destructans*.

### 3.3. Effects of PCA on Oxidative Stress, Energy Metabolism, and Apoptosis of P. destructans

Compared with the control, SOD and CAT activities showed no significant change at 20 μg/mL, increased significantly and peaked at 40 μg/mL, and declined at 50 μg/mL (Figure 2F,G). In line with these trends, DCFH-DA staining showed marked accumulation of intracellular ROS in treated hyphae (Figure 2I). Together with the SOD/CAT pattern, this indicates activation of antioxidant defenses in *P. destructans* that waned at the highest concentration. Meanwhile, intracellular ATP increased with concentration (Figure 2H), indicating an elevated energy state under PCA stress. Moreover, DAPI staining showed stronger nuclear fluorescence with chromatin condensation and nuclear fragmentation in treated hyphae, indicating DNA damage (Figure 2J). In Annexin V-FITC and PI dual staining, treated hyphae showed stronger green and red signals than the control, consistent with features of early and late apoptosis (Figure 2K).

### 3.4. Transcriptomic Analysis of P. destructans in Response to PCA Treatment

To evaluate PCA-induced transcriptomic changes, we performed RNA-seq on *P. destructans* from PCA-treated and control groups. Sequencing yielded 23 GB of raw data. After quality control, Q30 exceeded 93.73%, and alignment to the reference genome was above 89.09%, indicating high data quality (Appendix A). PCA showed clear separation between the two groups, indicating that PCA treatment altered gene expression patterns (Figure 3A). Differential expression analysis identified 1247 DEGs, with 966 downregulated and 281 upregulated (Figure 3B). GO enrichment showed that DEGs were concentrated in terms related to the cell cycle process, the mitotic cell cycle, DNA replication, the replication fork, and double-stranded DNA binding, suggesting that PCA suppresses growth by disturbing proliferation and genome replication in *P. destructans* (Figure 3C).

KEGG enrichment further indicated suppression of several core pathways, including Cell cycle—yeast (ko04111), Cell cycle (ko04110), and DNA replication (ko03030). DNA repair pathways—such as mismatch repair, homologous recombination, and nonhomologous end joining—were also downregulated (Figure 3D). GSEA supported the KEGG results: cell cycle and DNA replication pathways were broadly downregulated, whereas metabolic pathways such as oxidative phosphorylation (ko00190) tended to be upregulated, consistent with a PCA-induced stress response (Figure 3E). Network analysis identified five MCM subunits as central nodes, implying key roles in PCA-mediated interference with DNA replication and cell cycle progression (Figure 3F). Taken together, these data suggest that PCA inhibits *P. destructans* growth by modulating the cell cycle, DNA replication, and DNA repair. Detailed enrichment results are provided in Appendix A.

### 3.5. RT-qPCR Verification of RNA-Seq Data

To verify the transcriptomic results, we selected 12 representative differentially expressed genes (DEGs) for reverse transcription quantitative PCR (RT-qPCR). Expression trends matched the RNA-seq results, confirming the accuracy and reproducibility of the dataset (Appendix A).

### 3.6. Metabolomic Analysis of P. destructans in Response to PCA Treatment

To further probe PCA-induced metabolic responses, we performed untargeted metabolomics of *P. destructans* by UHPLC-MS/MS. OPLS-DA revealed clear separation between control and PCA-treated groups, indicating pronounced metabolic shifts in the mycelia (Figure 4A). Model metrics were R^2^Y = 0.996 and Q^2^ = 0.959, indicating a good fit and strong predictive ability (Figure 4B). In total, 1503 DEMs were identified, including 625 increases and 878 decreases.

By primary chemical class, DEMs fell into 21 categories, notably amino acids and their derivatives, organic acids and their derivatives, fatty acids, and glycerophospholipids (Figure 4C). KEGG enrichment indicated that several key pathways were significantly affected, especially those related to energy metabolism and cofactor synthesis. PCA significantly impacted cofactor biosynthesis, riboflavin metabolism, folate biosynthesis, and nucleotide sugar biosynthesis (*p* < 0.05). The DA scores for these pathways were negative, indicating an overall downward trend in the associated metabolic activities (Figure 4D). Detailed pathway information is provided in Appendix A. Overall, PCA markedly perturbed metabolism in *P. destructans*, broadly affecting amino acid metabolism, energy metabolism, and multiple biosynthetic routes.

### 3.7. Integration of Transcriptomic and Metabolomic Data

To gain deeper insight into how PCA acts on *P. destructans*, we integrated the transcriptomic and metabolomic datasets using O2PLS. Loadings from both datasets identified the top twenty genes and metabolites, including *HSP98* (VC83_08137), *HSP78* (VC83_00970), *PDE2* (VC83_00282), *FES1* (VC83_07843), and dihydroxyacetone phosphate (Figure 4E). In the joint analysis, these genes and metabolites varied in a highly coordinated manner along both axes (Figure 4F). Several pairs of metabolites and genes were significantly correlated (|r| > 0.8, *p* < 0.05); for example, propylparaben was negatively correlated with multiple genes, suggesting that these genes may participate directly or indirectly in its metabolic regulation (Figure 4G).

In addition, KEGG joint enrichment identified the 15 pathways with the largest numbers of enriched genes and metabolites (Appendix A), including glycerophospholipid metabolism, autophagy in yeast, ABC transporters, and purine metabolism (Figure 4H). These joint enrichments suggest that PCA may inhibit the growth of *P. destructans* by disturbing core processes, including membrane lipid metabolism, amino acid biosynthesis, stress responses, and energy metabolism.

## 4. Discussion

White-nose syndrome (WNS), caused by the psychrophilic fungus *Pseudogymnoascus destructans*, has caused steep declines in North American bat populations and is among the fungal infectious diseases with the highest mortality in nonhuman mammals [34]. Phenazine-1-carboxylic acid (PCA) is a natural antifungal metabolite produced by microbes, and prior studies have shown that it effectively inhibits *P. destructans* growth in vitro [7]. Here we systematically evaluated the in vitro antifungal activity of PCA and, together with physiological and biochemical assays, transcriptomics, and metabolomics, delineated its potential mechanisms at multiple levels. Our findings provide mechanistic evidence for the antifungal action of PCA and offer a conceptual basis for environmentally friendly management of WNS, supporting PCA as a promising candidate for practical use.

The fungal cell wall is essential for viability, morphology, and virulence, and is a principal target of antifungal agents. CFW staining showed that PCA treatment altered the cell wall staining pattern of *P. destructans* (Figure 2A), consistent with observations in *Pestalotiopsis kenyana* reported by Xun et al. (2023) [9]. We therefore examined cell wall polysaccharide metabolism and glycosylphosphatidylinositol (GPI) anchor biosynthesis in greater detail (Figure 5A). Transcriptome analysis showed downregulation of chitin synthase *CHS2_1* (VC83_02183) and β-1,3-glucan synthase *FKS1* (VC83_04653), suggesting impaired synthesis of chitin and 1,3-β-glucan. Meanwhile, expression of glycoside hydrolases, including chitinases, endoglucanases, and endomannanases, was variable, indicating that wall degradation processes were also affected. Metabolomics further confirmed decreases in multiple metabolites within the chitin, glucan, and mannan biosynthetic pathways. GPI is likewise critical for cell wall biosynthesis and organization; in *Aspergillus fumigatus*, GPI is indispensable for wall integrity, maintenance of morphology, and virulence [35]. Similar patterns have been observed in the response of *Colletotrichum gloeosporioides* to pterostilbene [36]. Taken together, the CFW staining patterns, downregulation of key biosynthetic genes, and depletion of chitin-, glucan- and mannan-related metabolites all point to impaired cell wall biosynthetic and remodeling processes in *P. destructans* after PCA treatment.

The fungal plasma membrane is composed mainly of lipids and proteins. It participates in signaling, cell recognition, and stress responses, and is a promising target for antifungal agents [37]. Omics data indicated that PCA treatment markedly perturbed membrane lipid metabolism. Specifically, the phospholipase C gene *plc* (VC83_01720) was upregulated, whereas the sphingolipid biosynthesis gene *SUR2* (VC83_05242) and the ergosterol biosynthesis gene *ERG1_2* (VC83_06307) were downregulated (Figure 5B). In addition, levels of key intermediates in glycerophospholipid and sphingolipid pathways, such as phosphocholine, acetylcholine, and ceramide, decreased significantly. Physiological and biochemical assays further corroborated membrane injury. PI staining revealed stronger red fluorescence in treated hyphae, indicating increased membrane permeability (Figure 2B). In addition, relative electrical conductivity increased in a time- and concentration-dependent manner, and extracellular nucleic acids and proteins increased in parallel, further supporting membrane damage (Figure 2C–E). Prior work likewise showed that PCA disrupts the membrane structure of *Pseudopestalotiopsis camelliae-sinensis*, consistent with our findings [38]. Notably, membrane disruption may impair synthesis and maintenance of the cell wall because chitin and glucan production depends on enzyme complexes located in the membrane [39]. Moreover, hydrolysis of glycerophospholipids can trigger apoptosis, and Annexin V-FITC and PI staining indicated that PCA induces programmed cell death in *P. destructans* [40,41] (Figure 2K).

In *P. destructans*, multiple putative virulence genes were differentially expressed after PCA treatment (Appendix A). Transcriptome analysis showed downregulation of the secreted protease *SP3* (VC83_09074). The encoded subtilase family protease is regarded as a major invasive factor during WNS and is associated with host tissue damage and skin necrosis [42]. Moreover, Reeder et al. (2017) reported that, during WNS, *P. destructans* upregulates virulence genes involved in the heat shock response, cell wall remodeling, and trace element acquisition; in our study, these genes were generally downregulated [43]. Integrated network analysis identified the heat shock proteins *HSP98* and *HSP78* as key nodes that were downregulated after PCA treatment (Figure 4G). As molecular chaperones, HSPs are central to protein folding, proteostasis, and the regulation of fungal virulence. Loss of *HSP70* reduces virulence in *Cryptococcus neoformans*, whereas overexpression of *HSP90* and *HSP70* in *Candida albicans* promotes proper folding and trafficking of virulence factors [44,45]. In addition, suppression of HSP expression can induce apoptosis, consistent with our physiological assays (Figure 2K) [46]. Metabolomics showed enrichment of the riboflavin pathway with a DA score of −0.5, indicating reduced biosynthesis (Figure 4D). Riboflavin is an established virulence metabolite in *P. destructans*; its accumulation at infection sites can disrupt host cell function and induce necrosis [47]. Taken together, PCA attenuates the pathogenic capacity of *P. destructans* by suppressing core virulence pathways—including secreted proteases, cell wall remodeling, ion homeostasis, and the heat shock response—and by interfering with riboflavin synthesis.

Mitochondria are central to energy metabolism in eukaryotes. They drive ATP synthesis through the TCA cycle and oxidative phosphorylation [48]. GSEA indicated upregulation of the oxidative phosphorylation pathway. Key genes, including *SDH2* (VC83_03131), *KGD2* (VC83_04627), and *COX17* (VC83_04361), were upregulated, suggesting that, after PCA treatment, *P. destructans* may sustain energy supply by boosting mitochondrial respiratory chain activity (Figure 3E). This mirrors compensatory energy metabolism observed when *Bacillus subtilis* Z-14 inhibits *Fusarium oxysporum* [49]. However, metabolomics showed an overall decrease in TCA intermediates (DA score = −0.625) alongside glucose accumulation, while biochemical assays also showed increased ATP (Figure 2H). These findings suggest coordinated use of glycolysis and oxidative phosphorylation to compensate for limited TCA substrates, a pattern also reported in *C. albicans* [50]. The respiratory chain can generate ROS [51]. Consistent with this, DCFH DA staining showed intracellular ROS accumulation after PCA treatment (Figure 2I). Excess ROS causes oxidative damage to DNA, proteins, and lipids, leading to apoptosis [52]. Annexin V-FITC and PI together with DAPI staining confirmed DNA damage and apoptosis in *P. destructans* after PCA treatment (Figure 2J,K), consistent with findings in *Botrytis cinerea* treated with 2-phenylethanol [53]. To mitigate oxidative stress, fungi typically activate enzymatic and nonenzymatic antioxidant systems. SOD and CAT form the first line of defense [52] and showed increased activity after PCA treatment, indicating activation of antioxidant defenses (Figure 2F,G). Glutathione is a major nonenzymatic scavenger [54]. Metabolomics showed a marked increase in oxidized glutathione (GSSG) and a decrease in reduced glutathione (GSH), indicating substantial consumption during ROS clearance. In addition, the pentose phosphate pathway was suppressed (DA score = −0.3), which may limit NADPH supply and thereby impede reduction of GSSG to GSH, weakening the glutathione cycle. Together, PCA likely enhances oxidative phosphorylation in *P. destructans*, drives ROS accumulation with compensatory activation of antioxidant systems, disrupts intracellular redox balance, and ultimately promotes apoptosis. While the omics and biochemical evidence strongly support a model of respiratory chain hyperactivation, future studies employing direct measurements of mitochondrial membrane potential or oxygen consumption rates are warranted to definitively validate this physiological state.

The cell cycle is central to morphogenesis and virulence regulation in fungi. Its disruption markedly suppresses growth and pathogenicity [55]. GSEA indicated broad downregulation of the cell cycle and DNA replication pathways after PCA treatment (Figure 3E). The eukaryotic cell cycle comprises four phases: G1, S, G2, and M [56]. Transcriptome data showed downregulation of several licensing factors for replication initiation, including *ORC5*, *MCM2-MCM6*, and *CDC45* (Figure 6). Initiation of genome replication depends on assembly and activation of the pre-replication complex [57]. Network analysis further identified multiple MCM subunits as central nodes (Figure 3F), consistent with their core role in replication licensing. During G1, the ORC–CDC6 complex loads MCM2-MCM7 onto replication origins to form the pre-replication complex. When MCM is insufficient, DNA replication fails to initiate on time and the G1 to S transition is impeded [58]. The coordinated downregulation of these genes suggests impaired replication licensing and a potential trigger for early replication stress.

During S phase, pathways for DNA replication elongation and DNA repair were broadly downregulated. Core factors were downregulated, including subunits of the DNA polymerase complex (*POL1*, *POL12*, *POLD4*, *DPB2*), RPA and RFC, and components of mismatch repair (*MSH2*, *MSH3*, *MSH6*), homologous recombination (*RAD51*, *RAD54*), and nonhomologous end joining (*POL4*, *KU70/80*). These patterns indicate a coordinated loss of replication fork stability and multiple repair capacities. Broad suppression of these replication and repair factors can stall replication forks, promote DNA damage accumulation, and cause genome instability [59]. Consistent with this, DAPI staining showed DNA damage (Figure 2J), providing phenotypic evidence of genomic injury. A similar mechanism was reported by Fan et al. (2024): treatment of *B. cinerea* with epinecidin-1 broadly downregulated genes for replication elongation and replication factors [60]. In addition, downregulation of late regulators such as *CDC20* and *CDC14* is consistent with cell cycle arrest. In *C. albicans*, berberine exerts antifungal activity by inducing cell cycle arrest and DNA damage [61]. Taken together, PCA suppresses replication licensing, replication elongation, and multiple repair pathways, thereby weakening system-wide responses to replication and DNA damage, promoting DNA lesion accumulation, and disturbing cell cycle control. This mechanism acts together with cell wall and membrane damage, downregulation of virulence genes, and energy and redox imbalance to diminish the survival and pathogenicity of *P. destructans*.

Several previous studies have examined the antifungal activity of PCA in other microorganisms, including plant-pathogenic fungi and bacteria. In these systems, PCA has repeatedly been reported to induce intracellular ROS accumulation, disrupt antioxidant defenses, damage cell walls and membranes, and impair mitochondrial function, ultimately leading to growth inhibition or cell death [9,11,14,62]. Our results in *P. destructans* are broadly consistent with these findings: PCA treatment similarly increased ROS levels, altered SOD and CAT activities, enhanced membrane permeability and electrolyte leakage, and caused apoptosis. At the same time, our integrative transcriptomic and metabolomic analyses in *P. destructans* reveal particularly pronounced suppression of DNA replication and repair pathways, cell-cycle regulation, and heat shock–dependent virulence networks, as well as interference with riboflavin biosynthesis. These features have not been systematically described for PCA in other species and suggest that, while PCA engages a set of conserved stress and damage responses across microorganisms, it also elicits species-specific vulnerabilities in this psychrophilic bat pathogen.

Notably, the EC_50_ of PCA against *P. destructans* determined in this study (approximately 32 μg/mL) indicates a moderate but biologically relevant level of antifungal activity. This value is higher than the MIC or EC_50_ values typically reported for highly optimized synthetic azole drugs such as fluconazole [63], and therefore PCA should not be considered comparable to such clinical agents in terms of intrinsic potency. Moreover, solubility, permeability and stability are inherent properties of PCA that will strongly influence the maximum effective concentration that can be achieved in vivo, and these physicochemical parameters were not quantitatively assessed in the present work. This represents one important limitation of our study. In addition, the transcriptomic and metabolomic analyses were performed at a single PCA concentration and at one time point with only a solvent control, so some of the observed changes may reflect general stress or growth inhibition rather than PCA-specific responses; this design also constitutes a major limitation of the present work. Unlike clinical pathogens for which drug resistance is routinely monitored, *P. destructans* is a psychrophilic pathogen of wild animals. For this species, the primary objective of PCA application is environmental management rather than systemic clinical therapy. Therefore, efficacy criteria for cave ecosystems must balance antifungal activity with environmental safety. Commercial fungicides typically show high environmental persistence and carry a risk of cross resistance among non-target fungal communities [64]. In contrast, PCA is a natural microbial metabolite and represents a potentially more biodegradable alternative. Although we did not include commercial reference compounds in the present experiments, which is a limitation of our study design, the pleiotropic mode of action revealed in this work suggests that PCA may impose selection pressure distinct from that of classical antifungal drugs. This difference may reduce the likelihood of rapid resistance development. At the same time, we did not assess non-target toxicity, PCA stability under hibernaculum-like conditions, or the reversibility of growth inhibition in this study, and these aspects will require dedicated investigation before any field application is considered.

Although this study provides a comprehensive characterization of the mechanisms by which PCA inhibits *P. destructans* in vitro, additional research is required to translate these laboratory findings into practical applications. First, in vivo and ex vivo evaluations using bat models and skin cell lines are essential to confirm the antifungal effectiveness of PCA and to assess its safety for host tissues and native cave microbiota. Second, although our omics analyses have identified potential candidates such as *SUR2* and *CHS2_1*, these candidates must be validated using genetic and biochemical approaches. Such experiments should include targeted gene deletion or overexpression, assays of direct binding or enzyme inhibition, and live cell measurements of mitochondrial membrane potential and respiratory activity. Third, systematic assessment of PCA solubility, permeability and stability, particularly under hibernaculum-like environmental conditions, will be necessary to define achievable exposure levels and to optimize formulation. Finally, given the distinctive environmental conditions in hibernacula, future work should focus on developing stable formulations and delivery strategies, such as aerosolized preparations, that maximize contact with the fungus while minimizing environmental impact, and on characterizing PCA persistence, degradation, and fungistatic versus fungicidal behavior using standard time–kill and washout assays. These efforts will be critical for advancing PCA from a promising candidate compound to a practical tool for WNS management.

## 5. Conclusions

Using physiological and biochemical assays, transcriptomics, and metabolomics, we systematically explored the inhibitory effects of PCA on *P. destructans* and its potential underlying mechanisms. The results confirm the in vitro inhibitory potential of PCA. PCA appears to affect both the cell wall and plasma membrane, modulate virulence factor expression, induce oxidative stress, perturb energy metabolism, and disrupt cell cycle progression and DNA replication and repair. These collective disruptions lead to DNA damage and apoptosis, suppressing fungal growth at multiple levels. The integrated evidence suggests that the antifungal activity of PCA is mediated through multifaceted disruptions of cellular homeostasis. Overall, this work expands the mechanistic basis for the antifungal use of PCA and highlights its potential utility as a candidate for environmentally friendly control of WNS.

## Figures and Tables

**Figure 1 jof-12-00016-f001:**
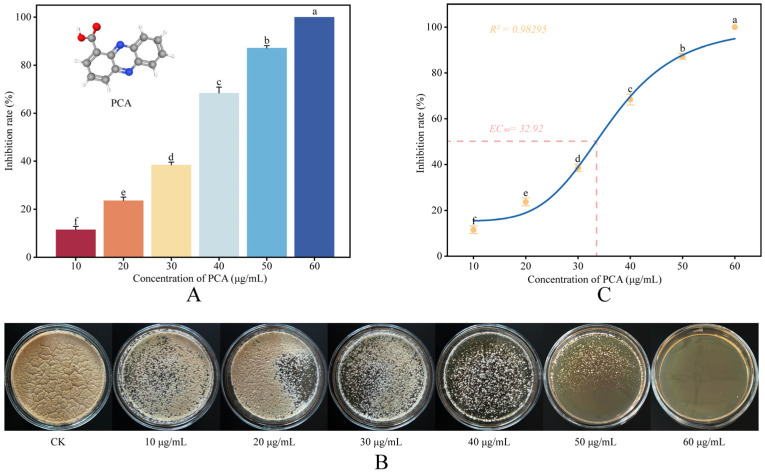
Antifungal activity of PCA against *P. destructans* in vitro. (**A**) Mycelial growth inhibition (%) of *P. destructans* at increasing concentrations of PCA. Data are presented as mean ± SD from three independent experiments (*n* = 3). Bars with different letters indicate significant differences between treatments (*p* < 0.05). (**B**) Representative colony morphology of *P. destructans* cultured with different PCA concentrations (0–60 μg/mL). (**C**) Concentration–response (dose–response) curve of PCA against *P. destructans*. Symbols represent mean ± SD (*n* = 3), and the fitted four-parameter logistic curve is shown. The calculated EC_50_ value is indicated on the graph. Points with different letters indicate significant differences between concentrations (*p* < 0.05).

**Figure 2 jof-12-00016-f002:**
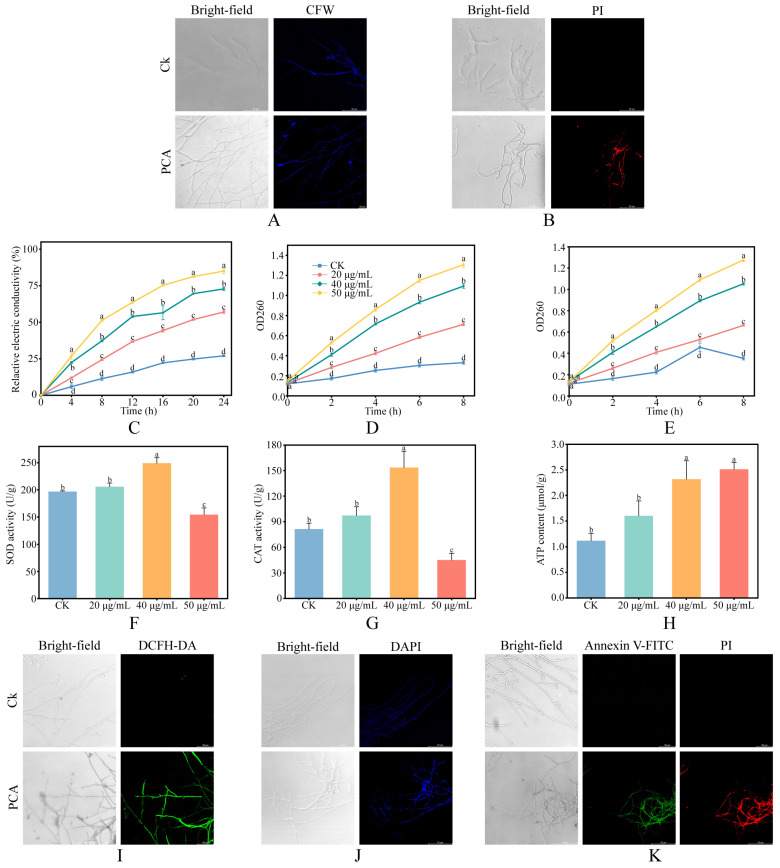
Effects of PCA on the phenotypes and key physiological and biochemical indices of *P. destructans*. (**A**) CFW staining; (**B**) PI staining; (**C**) time-course of relative electrical conductivity at 0/20/40/50 µg/mL; (**D**,**E**) leakage of nucleic acids (OD_260_) and proteins (OD_280_); (**F**–**H**) SOD and CAT activities and ATP content; (**I**) intracellular ROS by DCFH-DA; (**J**) nuclei by DAPI; (**K**) Annexin V-FITC/PI apoptosis assay. Magnification 400× ((**A**,**B**,**I**–**K**): bar 50 μm). Data are mean ± SD (*n* = 3); different letters indicate significant differences among treatments at the same time point or concentration (*p* < 0.05).

**Figure 3 jof-12-00016-f003:**
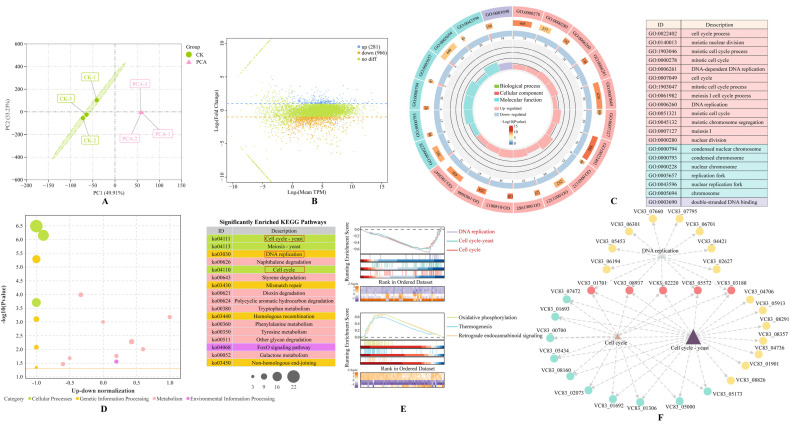
Transcriptomic analysis of *P. destructans* in response to PCA treatment. (**A**) Principal component analysis of gene expression in *P. destructans* treated with PCA. (**B**) MA plot of DEGs, with 966 genes downregulated and 281 upregulated. (**C**) Circular plot of GO enrichment analysis. The outer circle displays the top 20 GO terms; the middle circle indicates the number of background genes and the enrichment *p*-value; the inner circle represents the numbers of DEGs. (**D**) KEGG enrichment analysis of all significantly enriched pathways. (**E**) GSEA results based on KEGG pathway data. (**F**) Gene interaction network of selected KEGG pathways.

**Figure 4 jof-12-00016-f004:**
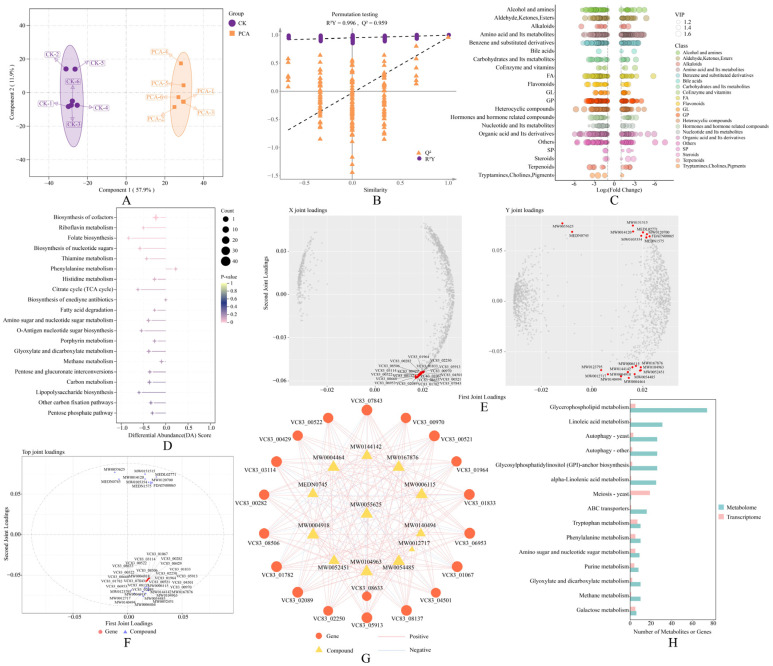
Metabolomic and integrated omics analysis of *P. destructans* under PCA treatment. (**A**) OPLS-DA score plot of metabolite profiles in CK and PCA-treated groups. (**B**) Permutation test validating the OPLS-DA model. (**C**) Level 1 classification bubble chart of DEMs. (**D**) DA score plot of the top 20 KEGG pathways enriched in DEMs. (**E**) O2PLS joint loading plot integrating transcriptomic and metabolomic datasets, highlighting the top 20 features. (**F**) Joint loading correlation plot showing the top 20 gene–metabolite associations from O2PLS analysis. (**G**) Correlation network of the top 20 metabolites and genes (|r| > 0.8, *p* < 0.05). (**H**) Bar chart of the top 15 KEGG pathways co-enriched in transcriptome and metabolome, indicating counts of genes and metabolites.

**Figure 5 jof-12-00016-f005:**
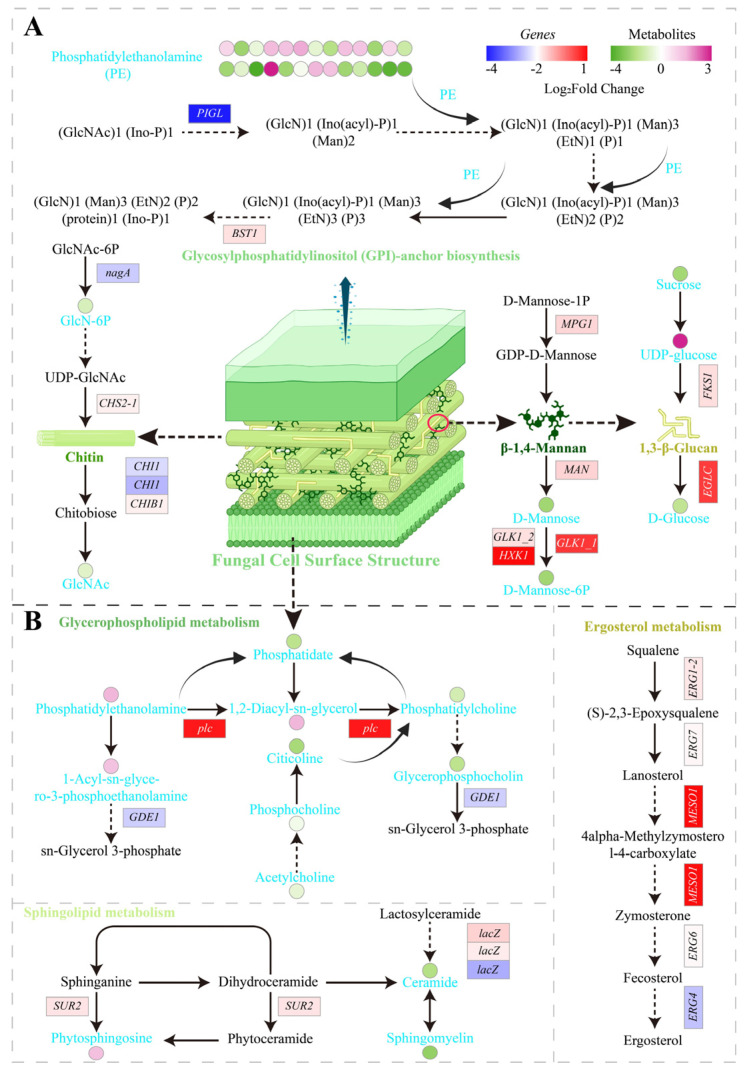
Effects of PCA on the cell wall and cell membrane of *P. destructans*. (**A**) Expression profiles of differentially expressed genes (DEGs, *n* = 3) and differentially expressed metabolites (DEMs, *n* = 6) associated with cell wall biosynthesis and metabolism. (**B**) Expression patterns of DEGs and DEMs involved in lipid and protein biosynthesis and metabolism related to the fungal cell membrane. Italicized labels indicate genes and metabolites. Rectangles represent genes, while circles denote metabolites.

**Figure 6 jof-12-00016-f006:**
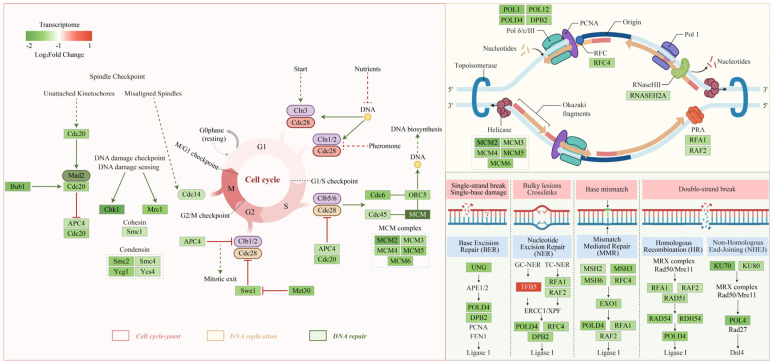
PCA suppresses cell-cycle progression, DNA replication, and DNA-repair pathways in *P. destructans*. Italicized labels indicate genes and metabolites. Rectangles represent genes, while circles denote metabolites.

## Data Availability

Transcriptome sequencing data were deposited into the NCBI Sequence Read Archive (SRA) under the accession number PRJNA1250769. Metabolome raw data are openly available in Figshare at https://doi.org/10.6084/m9.figshare.28794029.v1.

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
