# Peer review of "Multi-Omics Analyses Reveal the Antifungal Mechanism of Phenazine-1-Carboxylic Acid Against Pseudogymnoascus destructans"

_jof, 2025, doi:10.3390/jof12010016_

Round 1

Reviewer 1 Report

The topic is relevant and the application of multi-omics techniques to P. destructans is of potential interest, particularly given the ecological significance of this pathogen. While the study offers some degree of novelty by applying integrative omics to a fungus that has not been extensively profiled in this manner, I find that the current version of the manuscript requires substantial revision before it can be considered suitable for publication.

Although the authors generate sizeable transcriptomic, proteomic, and metabolomic datasets, most of the mechanistic claims are based solely on correlative omics changes and are not experimentally validated. The manuscript frequently asserts that PCA induces oxidative stress, compromises membrane integrity, and disrupts energy metabolism, but no direct experimental evidence is provided to support these conclusions. Without basic assays confirming ROS accumulation, membrane permeability, mitochondrial dysfunction, and targeted gene/protein expression, the mechanistic interpretation remains speculative.

In addition, the manuscript does not present essential antifungal phenotypic data that would contextualize the omics findings. Minimum inhibitory and fungicidal concentrations, dose-response behavior, time-kill dynamics, and morphological changes under PCA exposure are all missing, although these data form the foundation of any mechanistic antifungal study. Likewise, information regarding PCA such as stability in culture medium and appropriate solvent controls should be included to ensure that the biological observations can be interpreted reliably.

The proteomic and metabolomic components also require a more rigorous presentation of quality control and methodological transparency. Key parameters such as peptide coverage, FDR thresholds, sample clustering, internal standards, metabolite identification confidence levels, and enrichment statistics are insufficiently described. Without these details, it is difficult to assess the robustness and reproducibility of the datasets. The same concern applies to the transcriptomics, which would benefit from qRT-PCR validation of selected differentially expressed genes.

Beyond data-related issues, the manuscript would benefit from clearer methodological descriptions, a more focused introduction with a stronger statement of novelty, and revision of several sections for English language clarity. The conclusions should also be moderated, as the current version overstates the certainty of the proposed mechanism despite the lack of supporting functional assays.

The topic is relevant and the application of multi-omics techniques to P. destructans is of potential interest, particularly given the ecological significance of this pathogen. While the study offers some degree of novelty by applying integrative omics to a fungus that has not been extensively profiled in this manner, I find that the current version of the manuscript requires substantial revision before it can be considered suitable for publication.

Although the authors generate sizeable transcriptomic, proteomic, and metabolomic datasets, most of the mechanistic claims are based solely on correlative omics changes and are not experimentally validated. The manuscript frequently asserts that PCA induces oxidative stress, compromises membrane integrity, and disrupts energy metabolism, but no direct experimental evidence is provided to support these conclusions. Without basic assays confirming ROS accumulation, membrane permeability, mitochondrial dysfunction, and targeted gene/protein expression, the mechanistic interpretation remains speculative.

In addition, the manuscript does not present essential antifungal phenotypic data that would contextualize the omics findings. Minimum inhibitory and fungicidal concentrations, dose-response behavior, time-kill dynamics, and morphological changes under PCA exposure are all missing, although these data form the foundation of any mechanistic antifungal study. Likewise, information regarding PCA such as stability in culture medium and appropriate solvent controls should be included to ensure that the biological observations can be interpreted reliably.

The proteomic and metabolomic components also require a more rigorous presentation of quality control and methodological transparency. Key parameters such as peptide coverage, FDR thresholds, sample clustering, internal standards, metabolite identification confidence levels, and enrichment statistics are insufficiently described. Without these details, it is difficult to assess the robustness and reproducibility of the datasets. The same concern applies to the transcriptomics, which would benefit from qRT-PCR validation of selected differentially expressed genes.

Beyond data-related issues, the manuscript would benefit from clearer methodological descriptions, a more focused introduction with a stronger statement of novelty, and revision of several sections for English language clarity. The conclusions should also be moderated, as the current version overstates the certainty of the proposed mechanism despite the lack of supporting functional assays.

Reviewer 2 Report

Will be fine to haveone point Statistik in the text

also lets author put bars on picture on fig 2

separete fig 1 and 2 to be more clear for readers

No coments

Reviewer 3 Report

The manuscript presents the antifungal mechanism evaluation of phenazine-1-carboxylic acid against P. destructans responsible for the white-nose syndrome in bats.  In my opinion, the manuscript covers an interesting topic, the research is well-conducted and includes physiological, biochemical and multi-omics assays, along with molecular docking analyses and the paper is well-written. 

To enhance the value of this manuscript, authors can include future directions/ideas at the end of the discussion section.

The manuscript presents the antifungal mechanism evaluation of phenazine-1-carboxylic acid against P. destructans responsible for the white-nose syndrome in bats.  In my opinion, the manuscript covers an interesting topic, the research is well-conducted and includes physiological, biochemical and multi-omics assays, along with molecular docking analyses and the paper is well-written. 

To enhance the value of this manuscript, authors can include future directions/ideas at the end of the discussion section.

Reviewer 4 Report

Detailed comments

The manuscript explicitly hypothesizes several mechanisms of action for the compound studied, Phenazine-1-carboxylic acid (PCA), against Pseudogymnoascus destructans, based on multi-omics techniques and complementary analyses.

The authors enhance the study by incorporating various methodologies and seeking to link different fields of knowledge to address questions previously raised throughout the manuscript. Thus, the authors propose several mechanisms of action across the paper: damage to the cell wall and membrane, induction of oxidative stress, suppression of cell cycle, DNA replication and repair, interference with energy metabolism, and multiple targets that could be predicted from molecular docking showing distinct affinities of PCA for fungal proteins.

However, the methodologies present lack critical comparisons and have some limitations that introduce uncertainties into the results.

The biological activity method presented is non-standard: the agar dilution approach for quantifying mycelial growth is interesting, but it is not the standard method (broth microdilution). Although this methodology is recognized for filamentous fungi and allows for phenotypic assessment, it is less precise for determining low to intermediate activity. It is less comparable to results from commercial compounds using CLSI/EUCAST standards.

There are no comparisons with commercial antifungals. Therefore, there is no direct comparison of the results for PCA with other drugs tested under the same experimental conditions. There are no equivalent MIC/EC50 data for important reference compounds such as azoles, echinocandins, or other benchmark molecules. Even so, without any comparison, the authors portray the potential of the molecule as a strong activity. Since the IC50 for PCA (224.22 g/mol) was around 32 μg/mL, this could be considered moderate activity for a small molecule, but for fluconazole (306.271 g/mol) 32 μg/mL would be considered resistant. Thus, comparative activity benchmarking enables us to claim whether Treatment A or B is intriguing.

Additionally, the authors do not mention independent repetitions, duplicates, or triplicates. There is no cross-validation of data between alternative methods or complementary assays.

The effective concentration was obtained through colony area per plate, not cellular density, which may introduce quantification bias; what is the relationship between these methodologies? Is there a reference for this?

There is no reporting of toxicity against other eukaryotic cells (such as mammalian cells), molecular stability analysis, or inhibition/reversibility assays that are standard tests for validating new antifungal molecules.

The permeability assay using conductivity and leakage of nucleic acids/proteins is less common and relies on rigorous controls to avoid interference from the medium matrix. For example, PCA is a carboxylic acid, which means it can release H+ ions in aqueous solution and thus directly contribute to increased electrical conductivity of the medium proportional to its concentration. So, how can we be sure that the signal interpreted as leakage is not simply due to higher PCA concentration? Are there controls without cells and only PCA? And the inverse?

It is not demonstrated that all increases in conductivity are due to lysis/leakage of cellular components; there is no specific negative control.

Transcriptomic analysis points to strong downregulation of genes essential for cell wall biosynthesis, such as chitin synthase (CHS21) and glucan synthase (FKS1), indicating impaired production of key wall components. But could that simply be due to reduced growth (starvation)? How can we be sure that this downregulation is only due to the drug’s action? Controls?

Another point is that detailed electron micrographs of the fungus showing physical wall rupture are not shown, nor are there controls to rule out permeability changes confined only to the membrane without complete wall lysis.

Thus, there is an absence of controls that would allow the results to provide valuable insights into the mechanism of PCA.

Many of the statements made by the authors are based on results lacking controls or more thorough follow-up:

The text repeatedly claims that “PCA damages the cell wall of P. destructans,” “compromises the cell wall integrity,” and “likely disrupts the cell wall by suppressing biosynthesis while promoting the breakdown of wall polysaccharides and proteins.” However, the evidence for cell wall lysis is based only on changes in calcofluor white staining, reduced expression of wall biosynthesis genes, and leakage of intracellular content.

The manuscript makes claims such as “Docking analyses support a multi-target mode of action. SUR2 showed the highest predicted affinity for PCA, implicating disruption of sphingolipid homeostasis as a principal site of action,” and “These results suggest that PCA engages all nine candidate targets, supporting their potential roles as direct molecular mediators of PCA-induced fungal injury.” All evidence for molecular interactions is predictive, coming exclusively from molecular docking (in silico) and based on predicted binding energies. No direct binding assays, mutagenesis, altered protein expression, or functional tests were performed to validate these targets in live cells.

Finally, while the text at times qualifies findings (“we view these findings as supportive rather than definitive—target engagement and functional validation are still required”), the main conclusions are presented much more assertively than the experimental evidence allows.

Round 2

Reviewer 1 Report

I have carefully reviewed the revised version of the manuscript entitled “Multi-omics analyses reveal the antifungal mechanism of phenazine-1-carboxylic acid against Pseudogymnoascus destructans.” I would like to confirm that the authors have thoroughly addressed all of my previous comments and questions. The revisions have substantially improved the clarity and overall quality of the manuscript.

The responses provided were complete and satisfactory, and the additional data and explanations fully resolve the issues raised during the initial round of review. In my opinion, the manuscript now meets the journal’s standards in terms of methodology, interpretation of findings, and presentation.

Therefore, I am pleased to recommend the manuscript for acceptance in its current form.

I have carefully reviewed the revised version of the manuscript entitled “Multi-omics analyses reveal the antifungal mechanism of phenazine-1-carboxylic acid against Pseudogymnoascus destructans.” I would like to confirm that the authors have thoroughly addressed all of my previous comments and questions. The revisions have substantially improved the clarity and overall quality of the manuscript.

The responses provided were complete and satisfactory, and the additional data and explanations fully resolve the issues raised during the initial round of review. In my opinion, the manuscript now meets the journal’s standards in terms of methodology, interpretation of findings, and presentation.

Therefore, I am pleased to recommend the manuscript for acceptance in its current form.

Reviewer 4 Report

The authors confirmed that replicates were performed in their experiments, but there are still no standard deviation bars in Figure 1C. Wouldn't this affect the curve fitting (IC50)?

The concentration achieved by the manuscript's results is around 32 µg/mL. Even though the molecule is small (much smaller than fluconazole, for example), the authors still claim that the activity is strong. I disagree, since solubility and permeability are inherent to the molecule and can limit its action.

The effective concentration was obtained through the microorganism's area, and it is consistent with the references shown. My question concerns the different acquisition times for the areas, knowing that filamentous fungi grow radially. Can we say that the PCA in question is a growth inhibitor of the fungistatic type? Or do we call it fungicidal?

Regarding the conductometry test: The response is good at justifying the normalization, but it does not present numerical data or graphs (e.g., "water + PCA without cells over time" control) that would empirically demonstrate that PCA's direct contribution to conductivity is truly constant or negligible within the scale of the observed effect.​

It is implicit but not fully explicit whether it was verified that the boiling temperature and PCA presence do not differentially alter the L2 value between treatments, which could theoretically distort the relative conductivity.​

The absence of controls in the transcriptomics analysis reveals a manuscript limitation, where the authors do not rule out the possibility but also do not flag it as a major study limitation.​

In my view, the discussion needs consistent improvements; there are many references that used the microorganism and PCA—why not compare them?

Nothing

Round 3

Reviewer 4 Report

Nothing

Nothing

Author Response

We would like to express our sincere gratitude to the reviewer for the time and effort dedicated to reviewing our manuscript. We are pleased that our previous revisions addressed your concerns and that you have no further comments. Thank you once again for your professional support and positive feedback throughout the process.